# Transition-metal-free formal cross-coupling of aryl methyl sulfoxides and alcohols via nucleophilic activation of C-S bond

Guolin Li[1,2,4], Yexenia Nieves-Quinones[3,4], Hui Zhang[1], Qingjin Liang[1], Shuaisong Su[1], Qingchao Liu[2], Marisa C. Kozlowski [3✉] & Tiezheng Jia [1✉]

Employment of sulfoxides as electrophiles in cross-coupling reactions remains under-explored. Herein we report a transition-metal-free cross-coupling strategy utilizing aryl (heteroaryl) methyl sulfoxides and alcohols to afford alkyl aryl(heteroaryl) ethers. Two drug molecules were successfully prepared using this protocol as a key step, emphasizing its potential utility in medicinal chemistry. A DFT computational study suggests that the reaction proceeds via initial addition of the alkoxide to the sulfoxide. This adduct facilitates further intramolecular addition of the alkoxide to the aromatic ring wherein charge on the aromatic system is stabilized by the nearby potassium cation. Rate-determining fragmentation then delivers methyl sulfenate and the aryl or heteroaryl ether. This study establishes the feasibility of nucleophilic addition to an appended sulfoxide as a means to form a bond to aryl (heteroaryl) systems and this modality is expected to find use with many other electrophiles and nucleophiles leading to new cross-coupling processes.

[1] Shenzhen Grubbs Institute, Department of Chemistry and Guangdong Provincial Key Laboratory of Catalysis, Southern University of Science and Technology, Shenzhen, Guangdong 518055, China. [2] College of Chemical Engineering, Department of Pharmaceutical Engineering, Northwest University, Taibai North Road 229, Xi'an, Shaanxi 71009, China. [3] Roy and Diana Vagelos Laboratories, Penn/Merck Laboratory for High-Throughput Experimentation, Department of Chemistry, University of Pennsylvania, 231 South 34th Street, Philadelphia, PA 19104-6323, USA. [4] These authors contributed equally: Guolin Li, Yexenia Nieves-Quinones. ✉email: marisa@sas.upenn.edu; jiatz@sustech.edu.cn

Though cross-coupling reactions have become tremendously enabling in the past decades, further expansion of the electrophilic partners beyond conventional aryl halides, sulfonates, or carbonates, remains a considerable challenge[1–10]. The use of organosulfur compounds in cross-coupling is a new horizon that merits study due to their availability, chemical robustness, and structural versatility[11–21]. To date, a number of cross-coupling reactions utilizing aryl sulfides or sulfones have been reported, as exemplified by the well-known Liebeskind-Srogl cross-coupling reaction[16,19,20,22]. In sharp contrast, employment of sulfoxides as electrophiles in transition-metal catalyzed cross-coupling reactions via C-S bond activation remains rare (Fig. 1).

In 1979, Wenkert and coworkers pioneered the first nickel-catalyzed Kumada coupling reaction employing diaryl sulfoxides as electrophiles (Fig. 1a)[23]. In 2017, the Yorimitsu group reported a palladium-catalyzed Sonogashira coupling employing diaryl sulfoxides as electrophilic partners (Fig. 1b)[24]. The same group also described a second generation Buchwald-type palladacycle precatalyst that facilitated the double borylation of diaryl sulfoxides with diborane species (Fig. 1c)[25]. Both of the protocols shared some drawbacks, primarily low yields and limited functional group tolerance. Very recently, the same group reported a Buchwald-Hartwig amination between diaryl sulfoxides and amines catalyzed by an N-heterocyclic carbene-ligated palladium complex (Pd/SingaCycle-A1) (Fig. 1d)[26]. They noticed that aryl methyl sulfoxides, unfortunately, suffered from very low yields. The first systematic study employing alkyl aryl sulfoxides as electrophiles emerged in 2017 from the Yorimitsu group[27]. They demonstrated that synthetically accessible aryl methyl sulfoxides could undergo nickel-catalyzed oxidative addition of C–S bond, which was compatible with a Negishi coupling procedure to afford biaryl products (Fig. 1e). However, considerable effort is needed to separate the homocoupling byproducts of the zinc reagents from the desired heterocoupling products. In reports of aryl sulfide cross-couplings, selected examples are also shown where the corresponding sulfoxides can act as the electrophilic partner[28–30].

In general, the previous transition-metal catalyzed cross-coupling reactions of sulfoxides with different nucleophiles rely on transition metal oxidative addition to the C–S bond[23–27]. Transition-metal catalysts can require complex ligands, which may add significantly to the overall costs. In addition, the removal of trace transition metals in pharmaceutical and other applications remains a challenge[31–34]. In contrast, transition-metal-free cross-coupling strategies can overcome many of these limitations[35,36]. Some Friedel-Crafts-type protocols fall into this category, but suffer from limited substrate scope (electron rich arenes) and limited control over the reactive position on a given aromatic substrate.

We envisioned that a neglected S-tetracovalent dialkoxyl alkyl aryl intermediate (Fig. 1f, see key intermediate), which was pioneered by Martin group several decades ago[37–39], could potentially be utilized to activate the C–S bond, facilitating alkyl aryl sulfoxides as electrophilic coupling partners in the subsequent transition-metal-free coupling reactions with alcohols. This strategy would allow selective conversion of thio or sulfoxide groups to alkoxy groups. Furthermore, the utilization of sulfoxides in coupling reactions represents an alternative and complimentary pathway to conventional partners, such as aryl (pseudo)halides or alcohols, since methyl sulfoxides are prepared from thiophenol derivatives, which are manufactured from benzenesulfonic acid or benzenesulfonyl chloride, independent of aryl (pseudo)halides or alcohols. Herein, we report a transition-metal-free cross-coupling strategy enabled by a base-substrate interaction mode and apply it to the reaction of aryl methyl sulfoxides with alcohols

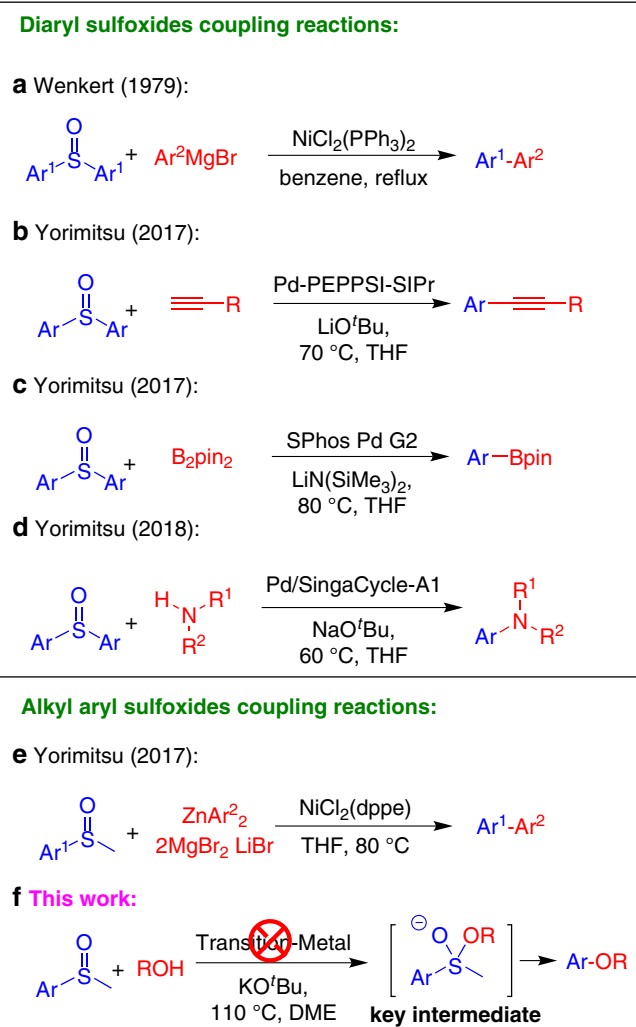

**Fig. 1 Cross-coupling reactions employing sulfoxides as electrophiles.** **a** Nickel-catalyzed Kumada coupling reaction of diaryl sulfoxides. **b** Palladium-catalyzed Sonogashira coupling reaction of diaryl sulfoxides. **c** Palladium-catalyzed borylation of diaryl sulfoxides. **d** Buchwald-Hartwig amination of diaryl sulfoxides and amines. **e** Nickel-catalyzed Negishi coupling reaction of aryl methyl sulfoxides. **f** This work: transition-metal-free formal cross-coupling of aryl methyl sulfoxides and alcohols.

to afford a general and complementary entry to alkyl aryl ethers (Fig. 1f).

## Results

**Optimization study.** Readily accessible methyl 2-naphthyl sulfoxide (**1a**) which possess no electronic or steric bias was chosen as the model substrate for the study (Table 1). The investigation was initialized by treating **1a** with three methoxide bases (LiOMe, NaOMe and KOMe) in 2-Me-THF at 110 °C for 12 h (Table 1, entries 1–3). Use of KOMe as base as well as the nucleophile afforded product **3a** in 12% yield. To improve reactivity, four different solvents [toluene, dimethoxyethane (DME), cyclopentyl methyl ether (CPME), and dioxane]) were examined using **1a** and KOMe (Table 1, entries 4–7). DME was the best solvent, generating **3a** in 33% yield (Table 1, entry 5). We next probed the roles of KOMe by using MeOH (**2a**) in the presence of different potassium bases [KO$^t$Bu, K$_2$CO$_3$, KOH, KN(SiMe$_3$)$_2$, Table 1, entries 8–11]. Notably, the yield of **3a** increased to 67% when KO$^t$Bu was employed as the base (Table 1, entry 8). Furthermore, the potential byproduct, 2-naphthyl tert-butyl ether, was observed

**Table 1 Optimization for transition-metal-free cross-coupling reaction between 1a and 2a[a].**

| Entry | Base/equiv | 2a/equiv | Solvent | Assay yield[b]/% |
|---|---|---|---|---|
| 1 | LiOMe/3.0 | — | 2-Me-THF | 0 |
| 2 | NaOMe/3.0 | — | 2-Me-THF | 2 |
| 3 | KOMe/3.0 | — | 2-Me-THF | 12 |
| 4 | KOMe/3.0 | — | toluene | 0 |
| 5 | KOMe/3.0 | — | DME | 33 |
| 6 | KOMe/3.0 | — | CPME | 7 |
| 7 | KOMe/3.0 | — | dioxane | 7 |
| 8 | KO$^t$Bu/3.0 | 3.0 | DME | 67 |
| 9 | K$_2$CO$_3$/3.0 | 3.0 | DME | 0 |
| 10 | KOH/3.0 | 3.0 | DME | 28 |
| 11 | KN(SiMe$_3$)$_2$/3.0 | 3.0 | DME | 10 |
| 12[c] | KO$^t$Bu/3.0 | 3.0 | DME | 95 |
| 13[d] | KO$^t$Bu/3.0 | 3.0 | DME | 91 |
| 14[c] | KO$^t$Bu/2.0 | 2.0 | DME | 94(90[e]) |
| 15[c] | KO$^t$Bu/2.0 | 1.5 | DME | 85 |
| 16[c] | KO$^t$Bu/1.5 | 2.0 | DME | 83 |
| 17[c,f] | KO$^t$Bu/2.0 | 2.0 | DME | 35 |

[a]Reaction conditions: **1a** (0.1 mmol), **2a** (0.3 mmol), base (0.3 mmol), solvent (1.0 mL) under argon atmosphere at 110 °C for 12 h.
[b]Assay yield determined by $^1$H NMR using 0.1 mmol (7.0 µL) CH$_2$Br$_2$ as internal standard.
[c]0.5 M concentration.
[d]1.0 M concentration.
[e]Isolated yield.
[f]80 °C.

only in trace amounts; presumably, the steric hindrance of O$^t$Bu limits its participation as a nucleophile. In the optimization, concentration was determined to be a key factor. When the concentration of the reaction was elevated to 0.5 M, product **3a** could be prepared in an optimal 95% yield (Table 1, entry 12). However, the yield of **3a** slightly dropped to 91% when the concentration was raised further to 1.0 M (Table 1, entry 13). The amounts of both base and nucleophile could be successfully decreased to 2 equivalents, and **3a** was still obtained in 94% assay yield, 90% isolated yield (Table 1, entry 14). Further decreasing the equivalents of either component or the temperature resulted in lower yields (Table 1, entries 15–17). Therefore, the optimal transition-metal-free cross-coupling reaction was 1 equivalent of **1a** as electrophile, 2 equivalents of methanol as nucleophile, and 2 equivalents of KO$^t$Bu as base in DME (0.5 M) at 110 °C for 12 h.

**Substrate scope**. Having identified optimal reaction conditions, we next explored the scope of substrates of aliphatic alcohols (Fig. 2). Methanol (**2a**) and ethanol (**2b**) were successfully utilized as coupling partners, generating **3a** and **3b** in 90% and 93% yield respectively. A secondary alcohol, isopropanol (**2c**), furnished the corresponding product **3c** in good yield (82%). Aliphatic alcohols possessing phenyl substituents either at the α- or γ- position were well tolerated, providing **3d** and **3e** in 61% and 86% yield, respectively. The reactions proceeded smoothly between **1a** and alcohols bearing cyclic substituents (**2f**, **2h**), affording the corresponding products in good to excellent yields. Even sterically hindered 1-adamantylmethanol (**2h**) could be utilized and produced **3h** in 80% yield. Moreover, alkyl alcohols with ether or amine functional groups were well tolerated and led to the formation of the desired products **3i-l** in excellent yields. Remarkably, the potentially competitive amination did not occur at all

with either primary amine **2k** or secondary amine **2l**. Our protocol could be utilized to functionalize more complex structures including natural products to afford **3m** and **3n** in excellent yields.

Next, the aryl group of the methyl sulfoxides was surveyed when cyclohexylethan-1-ol (**2g**) was employed as cross-coupling partner (Fig. 3). Phenyl methyl sulfoxide (**1b**) could successfully couple with **2g** to afford the corresponding **4a** in 76% yield under slightly modified conditions. Methyl sulfoxides bearing electron-withdrawing substituents such as cyano or trifluoromethyl groups underwent the cross-coupling reaction to afford compounds **4b** and **4c** in excellent yields. Substrates with electron-donating groups reacted more sluggishly; even so, **4d** and **4e** were successfully prepared using our method in modest to good yields. It is noteworthy that the *para*-thiomethyl group barely reacted, in contrast to prior reports involving transition metal catalysts[23,28–30], allowing the formation of **4e** and highlighting the unique chemoselectivity of our protocol. Steric hindrance on the aryl sulfoxide portion exerted negligible influence on our protocol as evidenced by the 1-naphthyl derivative **4f** forming in 94% yield. Methyl sulfoxides possessing a polymerizable vinyl group (**1h**) or and electrophilic carbonyl group (**1i**) were also compatible, albeit giving **4g** and **4h** in modest yields. Alkyl heteroaryl ethers are ubiquitous in biologically relevant compounds[40]. Remarkably, a range of heteroaryl groups exhibited good to excellent compatibility under the optimal conditions; pyridyl (**4i**, **4j**, **4k**), isoquinolyl (**4l**), imidazolyl (**4m**), pyrimidinyl (**4n**), benzoimidazolyl (**4o**) as well as quinolyl (**4p-t**) ethers could be obtained in 66–99% yields. A small number of pyridyl sulfoxides have been found to undergo substitution with alkoxides, but reduction was an accompanying product[41,42]. Of note, bromo- or iodo-substituents on **4r-t** were compatible with this method, providing a means for later modification via orthogonal transition-metal

**Fig. 2 Substrate scope alcohols with 1a.** All reactions were conducted with **1a** (0.2 mmol), **2** (0.4 mmol), KO*t*Bu (0.4 mmol) in DME (0.4 mL) under argon atmosphere at 110 °C for 12 h.

catalyzed coupling strategies. Significantly, this protocol could even be utilized to directly derivatize 2-(methylsulfinyl)-4-(2-thiophenyl)-6- (trifluoromethyl)pyrimidine (**1v**), a patented compound with anti-apoptosis bioactivity[43], to deliver **4u** in 72% yield. To establish the scalability of our transition-metal-free coupling procedure, a gram-scale reaction with **1q** (5.5 mmol, 1.05 g) and **2g** (11.0 mmol, 1.41 g) was performed, affording **4p** in excellent yield (99%, 1.48 g).

**Synthetic applications**. To illustrate the potential utility of a transition-metal free coupling reaction of aryl methyl sulfoxides with alcohols in medicinal chemistry and drug development, two drug candidates have been prepared employing this method as the key step (Fig. 4). A sphingosine 1-phosphate receptor modulator **6e**, which is currently in phase II clinical trials for the treatment of multiple sclerosis[44], could be synthesized in 5 steps and 20% overall yield. Notably, no transition metals are utilized in the entire route, simplifying the drug purification process. Another patented compound, opioid delta receptor agonist **7c**[45], could be generated in only 3 steps and 61% overall yield using our highly chemoselective transition-metal free coupling strategy as the key step.

From the standpoint of atom economy, aryl methyl sulfoxides are the most appealing of sulfoxide coupling partners. Nevertheless, other alkyl aryl sulfoxides were also explored (Table 2). Ethyl 2-naphtyl sulfoxide (**5a**) proved to be a suitable coupling partner, despite affording **3a** in lower yield (Table 2, **5a**). As expected, the other three alkyl sulfoxides under investigation (**5b–d**) only generated trace amounts of **3a**, presumably owing to rapid

formation of sulfenate anions after attack of methoxide on the sulfur[46–48] (Table 2, **5b–5d**).

To understand the mechanism, a series of control experiments were performed (Table 3). When 2.0 equiv of TEMPO was introduced under the standard reaction conditions, the yield of **3a** was not significantly affected (see Supplementary Information for details), which points away from a radical pathway. Next, KO*t*Bu was replaced by two other *tert*-butoxide bases (LiO*t*Bu, NaO*t*Bu) in the transition-metal-free cross-coupling protocol, with the rest of the conditions kept the same. Despite the similar basicity of these reagents, the yields of **3a** were dramatically decreased, which illustrates the crucial role of the potassium cation (Table 3, entries 1–2). To further confirm the effect, 2.0 equivalents 18-crown-6 were introduced into the otherwise standard conditions, and the yield of **3a** dropped precipitously to 5% (Table 3, entry 3). In a subsequent study, 2.0 equivalents of potassium fluoride added to the reactions with LiO*t*Bu or NaO*t*Bu restored the reactivity providing 63% and 27% yield, respectively (Table 3, entries 4–5). These control experiments support a pivotal role of potassium cation in the transition-metal-free cross-coupling[49,50]. Hence, a computational study was conducted to fully understand this unusual C-S bond activation mode for methyl sulfoxides that does not require a transition metal catalyst.

**Mechanistic studies**. Computational studies focused on the reaction of **1a** with methoxide using different cations (K, Na and Li). DFT calculations were performed using Gaussian 09. All geometry optimizations and vibrational frequency calculations of stationary points and transition states (TSs) were carried out at

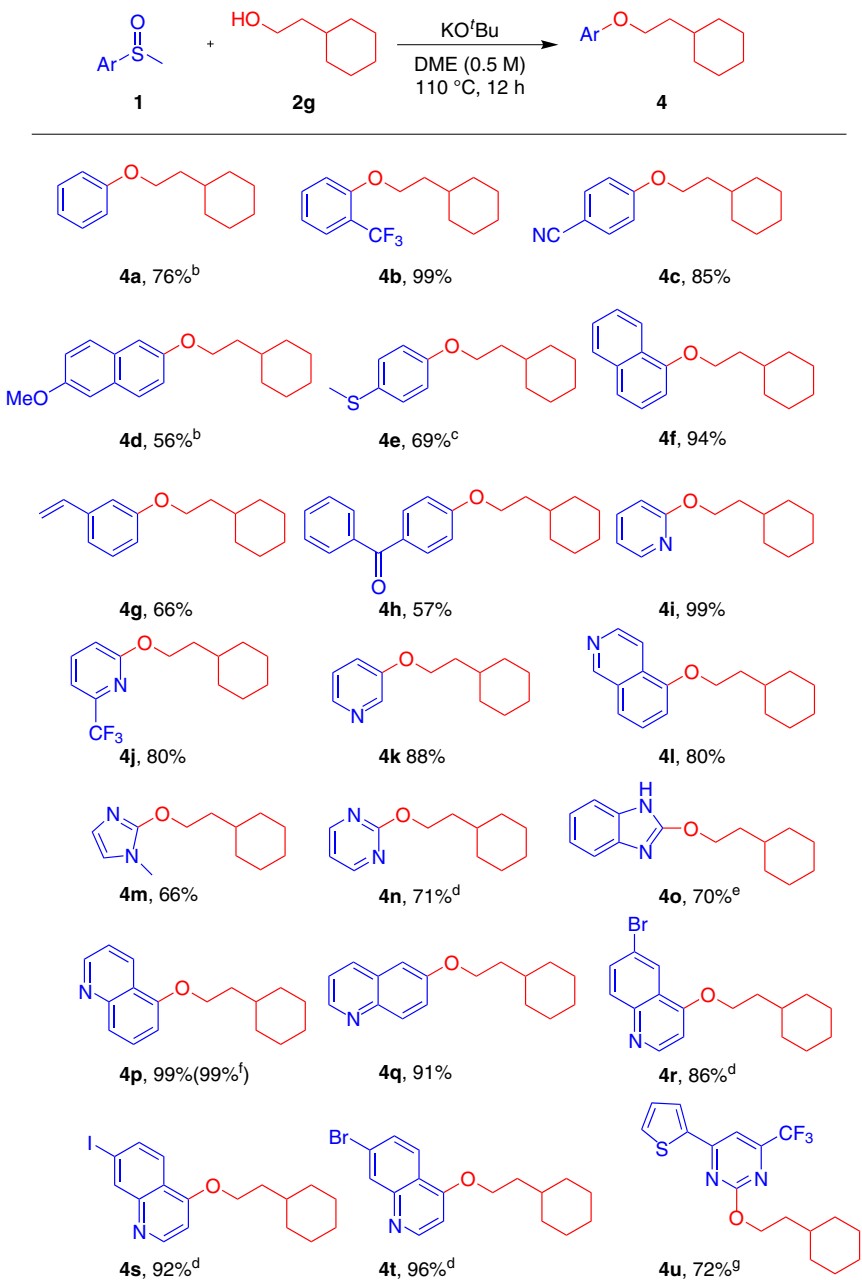

**Fig. 3 Substrate scope of aryl methyl sulfoxides with 2g[a].** [a]Unless noted, all reactions were conducted with **1** (0.2 mmol), **2 g** (0.4 mmol), KO$^t$Bu (0.4 mmol) in DME (0.4 mL) under argon atmosphere at 110 °C for 12 h. [b]**2g** (3.0 equiv), KO$^t$Bu (3.0 equiv) at 110 °C for 24 h. [c]**2g** (3.0 equiv), KO$^t$Bu (3.0 equiv). [d]40 °C; [e]**2g** (3.0 equiv), KO$^t$Bu (4.0 equiv) at 100 °C for 30 h. [f]Large scale reaction: **1q** (5.5 mmol), **2g** (11.0 mmol), KO$^t$Bu (11.0 mmol) in DME (11.0 mL) under argon atmosphere at 110 °C for 12 h. [g]**1u** (4.0 equiv), **2g** (1.0 equiv), KO$^t$Bu (2.0 equiv) at 40 °C for 48 h.

the B3LYP level of theory with 6-31+G(d) basis set for all atoms. Single point energies and solvent effects in 1,4-dioxane were computed at the M06-2X level of theory with 6-31+G(d,p) basis set for all atoms, using the gas-phase optimized structures. Solvation energies were calculated by a self-consistent reaction field (SCRF) using the SMD solvation model.

Several possible mechanisms were taken into consideration (Fig. 5). The first proposal (Fig. 5a), involves a S$_N$Ar reaction between the aryl methyl sulfoxide and the alkoxide, initiated by a π-cation interaction between the potassium and the aromatic ring[51–53]. A second proposal (Fig. 5b) proceeds via concerted addition of the alkoxide and elimination of the methyl sulfoxide group. The third mechanistic possibility involves nucleophilic addition of the alkoxide to the methyl sulfoxide facilitated by

coordination of the potassium cation to the aromatic ring and the oxygen from the sulfoxide group. The resultant intermediate can subsequently undergo cyclization via attack of the oxygen atom on the ipso carbon. The final step entails the elimination of the methyl sulfenate to afford the desired product (Fig. 5c).

The first mechanistic proposal was quickly found to be untenable due to far more favorable coordination of the cation to the sulfoxide oxygen rather than via a π-cation complex. Intermediates and transition states for the second mechanistic possibility were located. However, the computational energies indicate that the potassium counterion should decrease the reactivity relative to sodium and lithium (see Supplementary Fig. 1), which is the opposite of what was observed experimentally.

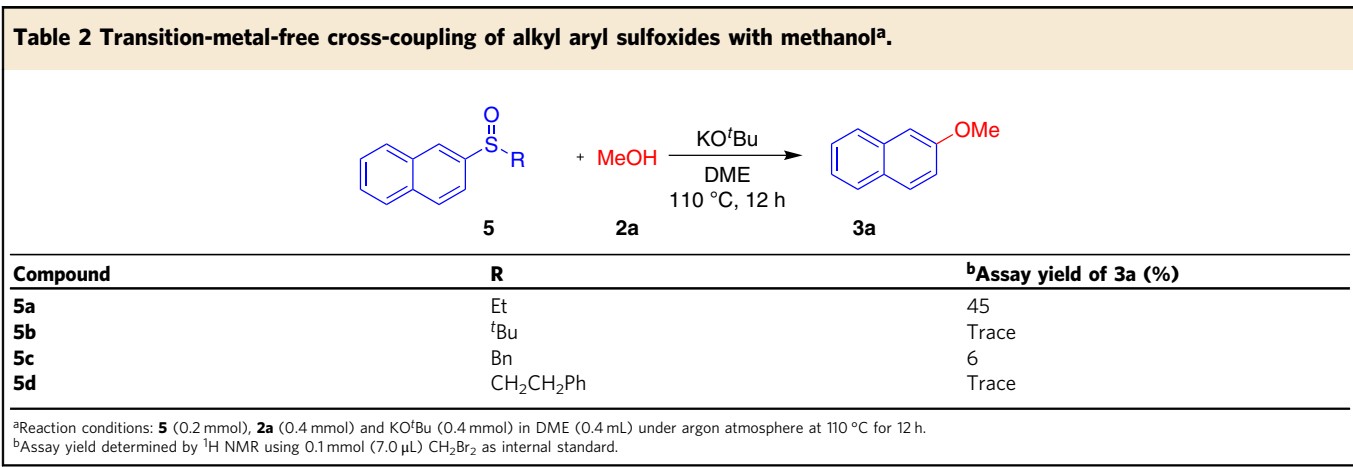

**Fig. 4 Synthetic applications.** Application of transition-metal-free cross-coupling of aryl methyl sulfoxides with alcohols to pharmaceuticals.

**Table 2 Transition-metal-free cross-coupling of alkyl aryl sulfoxides with methanol[a].**

| Compound | R | [b]Assay yield of 3a (%) |
|---|---|---|
| 5a | Et | 45 |
| 5b | $^t$Bu | Trace |
| 5c | Bn | 6 |
| 5d | CH$_2$CH$_2$Ph | Trace |

[a]Reaction conditions: **5** (0.2 mmol), **2a** (0.4 mmol) and KO$^t$Bu (0.4 mmol) in DME (0.4 mL) under argon atmosphere at 110 °C for 12 h.
[b]Assay yield determined by $^1$H NMR using 0.1 mmol (7.0 μL) CH$_2$Br$_2$ as internal standard.

In contrast, the third mechanism (Fig. 5c) accounts for the experimentally observed cation effect (Fig. 6). Calculations suggest that the reaction proceeds via a nucleophilic attack of the alkoxide to the sulfoxide, which will lead to coordination of the counterion with the C1 carbon and formation of a three membered ring via attack of the alkoxy oxygen to the ipso carbon of **1a** to afford **INT1**. This intermediate can then undergo an elimination step via **TS1** to afford the desired product and methyl sulfenate. The energy barrier to obtain the desired product is significantly lower for potassium than for sodium or lithium. Calculations suggest that the nature of the cation will affect both the stability of **INT1** and the subsequent elimination of the methyl sulfenate. Experimental observations showed that the use of LiO$^t$Bu completely inhibits reactivity, and that NaO$^t$Bu significantly decreases the yield (Table 3, entries 1–2), further supporting the computational findings. Since this mechanism relies on initial nucleophilic attack of the sulfoxide, it also consistent with more electrophilic substrates reacting more quickly (Table 2, **5b** and **5c**) than those with electron donating groups (Table 2, **5d**). For heterocycles with a nitrogen adjacent to the sulfoxide (**4i**, **4j**, **4m**, **4n**, **4o**, **4u**), reactivity may be further assisted by bidentate coordination of the potassium to the heterocyclic nitrogen and the sulfoxide oxygen. However, the lack of any distinct trends in the outcomes for these substrates relative to heterocycles with more distal nitrogens (**4k**, **4l**, **4p**, **4q**, **4r**, **4s**, **4t**) indicates that this effect does not predominate.

**Table 3 Influences of bases and additives on the transition-metal-free cross-coupling[a].**

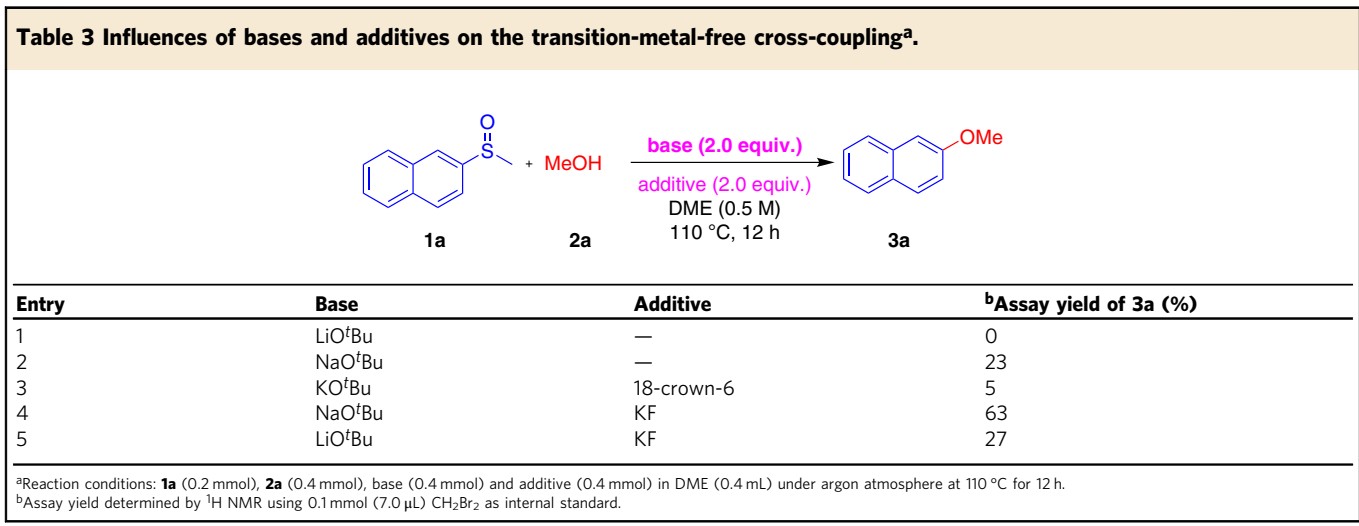

| Entry | Base | Additive | [b]Assay yield of 3a (%) |
|---|---|---|---|
| 1 | LiO$^t$Bu | — | 0 |
| 2 | NaO$^t$Bu | — | 23 |
| 3 | KO$^t$Bu | 18-crown-6 | 5 |
| 4 | NaO$^t$Bu | KF | 63 |
| 5 | LiO$^t$Bu | KF | 27 |

[a]Reaction conditions: **1a** (0.2 mmol), **2a** (0.4 mmol), base (0.4 mmol) and additive (0.4 mmol) in DME (0.4 mL) under argon atmosphere at 110 °C for 12 h.
[b]Assay yield determined by $^1$H NMR using 0.1 mmol (7.0 μL) CH$_2$Br$_2$ as internal standard.

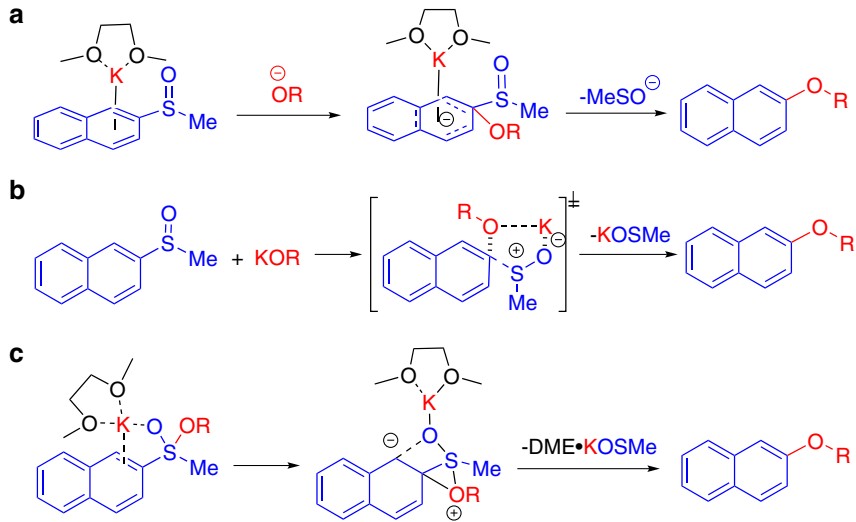

**Fig. 5 Possible mechanistic pathways.** The potassium assisted C–S bond activation of aryl methyl sulfoxides (**a-c**).

## Discussion

In summary, a unique transition-metal-free cross-coupling protocol utilizing aryl methyl sulfoxides as electrophilic partners with alcohols to afford alkyl aryl ethers is described. An array of functional groups is well tolerated, and the method can be used for late-stage functionalization of natural products and pharmaceuticals. The methyl sulfoxides utilized as coupling partners in our protocol were generally prepared by methylation of corresponding thiophenol derivatives, followed by oxidation by NaIO$_4$ or *m*-CPBA. Considering that thiophenol derivatives are typically manufactured by reduction of benzenesulfonic acid or benzenesulfonyl chloride, the method developed herein provides orthogonal entries relative to halides or alcohols, which permits different functional group compatibility and different substitution patterns. The successful application of the transition-metal free coupling reaction reported herein in drug syntheses and derivatization highlights its potential utility in medicinal chemistry as well as drug development. Both experimental results and a DFT-based computational study suggest that the reaction proceeds via a nucleophilic addition of the alkoxide to the sulfoxide accompanied by a cyclization. Subsequently, the alkoxide migrates to the ipso carbon while eliminating the methyl sulfenate. The dramatic impact of the cation on the reactivity arises both from

greater stabilization of **INT1** via association with the C1 carbon and enhanced ability to fragment via **TS1**. These results wherein an electrophilic partner is activated by addition of a nucleophile that both enhances its ability as leaving group and activates the aromatic substrate via coordination of a counterion open up possibilities for electrophile activation and coupling in general. The concepts described herein also provide a basis for the construction of systems to allow transformations of organosulfur compounds in other contexts.

## Methods

**General procedure for catalysis**. To an oven-dried microwave vial equipped with a stir bar was added KO$^t$Bu (44.9 mg, 0.4 mmol, 2 equiv), and 2-methanesulfinyl-naphthalene (38.0 mg, 0.2 mmol, 1 equiv) under argon atmosphere in a glove box. DME (0.4 mL) was added to the vial by syringe. The microwave vial was sealed and removed from the glove box. Then, methanol (16.2 μL, 0.4 mmol, 2 equiv) was added by syringe under argon atmosphere. Note that solid and viscous oil alcohols were added to the reaction vial prior to KO$^t$Bu. The reaction mixture was heated to 110 °C in an oil bath and stirred for 12 h. Upon completion of the reaction, the sealed vial was cooled to room temperature, and opened to air. The reaction mixture was passed through a short pad of silica gel. The pad was then rinsed with 10:1 dichloromethane:methanol. The resulting solution was subjected to reduced pressure to remove the volatile materials and yielded a viscous oil. The residue was purified by flash chromatography as outlined below.

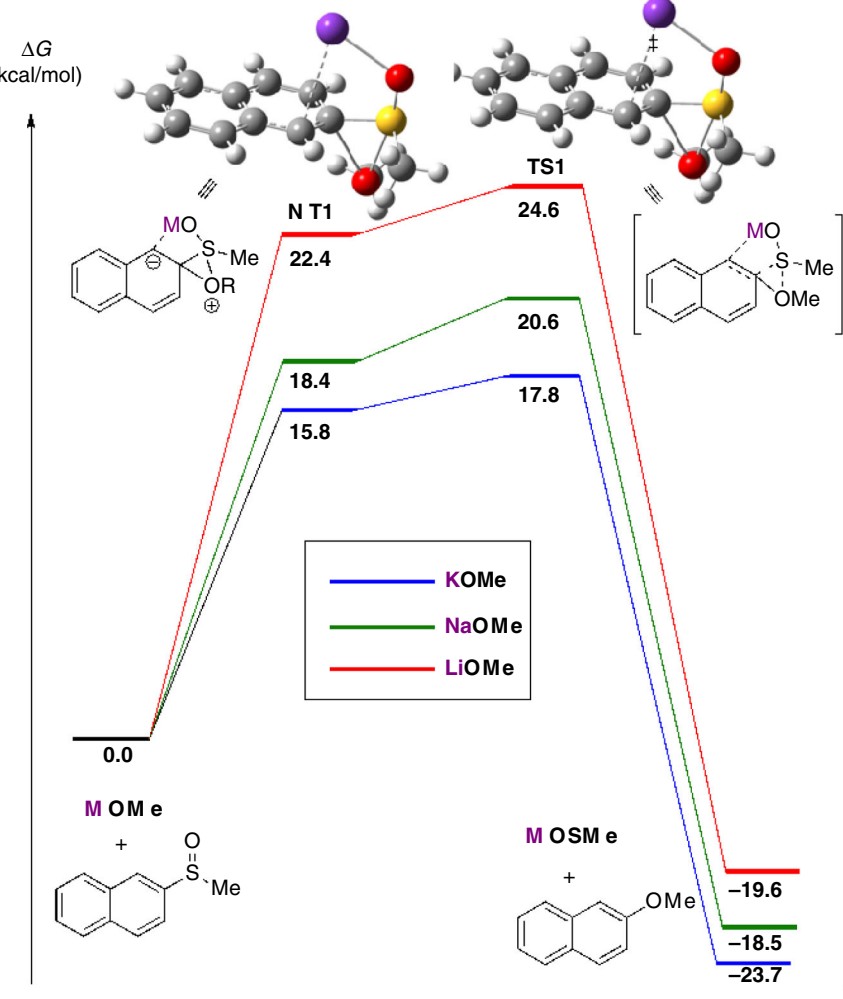

**Fig. 6 Free energy profile for the reaction of 1a with methoxide in presence of different counterions (M = Li, Na and K).** Relative free energy values were calculated with SMD-1,4-dioxane-M06-2×/6-31+G(d,p)// B3LYP/6-31+G(d).

## Data availability

Experimental procedure and characterization data of new compounds are available within the Supplementary Information. Any further relevant data are available from the authors upon reasonable request.

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

## Acknowledgements

T.J. thanks Shenzhen Nobel Prize Scientists Laboratory Project (C17783101), the Science and Technology Innovation Commission of Shenzhen Municipality (JCYJ20180302180256215), and Guangdong Provincial Key Laboratory of Catalysis (2020B121201002) for financial support. SUSTech is gratefully acknowledged for providing startup funds to T.J. (Y01216129). M.C.K. thanks the NIH (GM131902) for financial support and XSEDE (TG-CHE120052) for computational support.

## Author contributions

G.L., Q.L., H.Z., Q.L. and S.S. performed the experiments. Y.N.-O. carried out computational study. M.C.K. directed the part of computational study. T.J. and M.C.K. conceived and directed the project and wrote this paper. All authors approved the submission of the manuscript. G.L. and Y.N.-O. contributed equally.

## Competing interests

The authors declare no competing interests.
