## [Peer Review File · Nature Communications]

Reviewers' comments:

Reviewer #1 (Remarks to the Author):

Aryl sulfides could act as nucleophiles in transition-metal-catalyzed cross coupling reactions, such as Liebeskind-Srogl reactions (ref. 22-25 in the manuscript), which usually proceed via two-electron transition-metal addition pathway. Aryl sulfones, on the other hand, could be good radical precursors, as shown in Julia fragmentation. Therefore, it is a very interesting question to probe the role of aryl sulfoxides in coupling reactions. It is intriguing to a broad range of chemists that aryl sulfoxides could react in a completely different pathway, especially independent of transition-metal catalysts. Jia, Kozlowski and coworkers developed an amazing cross-coupling reaction between aryl sulfoxides and alcohols. Remarkably, the protocol performed well undergo transition-metal-free manner in the presence of K⁺O⁻tBu. Thus I am both excited and intrigued by this unprecedented and novel transformation. Moreover, the substrate scope is well demonstrated, and the compounds are well characterized. They also applied their method in the preparation of two bioactive molecules to demonstrate its synthetic potential. The novel mechanism disclosed in the manuscript, which was demonstrated by DFT calculation, certainly could lead to further developments. Jia and Kozlowski's hypothesis-driven research style as shown in the manuscript were inspired by JC Martin's work in the 70th, and applied the key intermediate in the modern organic methodology. Therefore, I recommend the acceptance of this manuscript to Nature Communications after address the following notes:

1. In manuscript, in conclusion part, it should be "late-stage", not "late-state".
2. In SI, calculated value of compound 3h for HRMS should have 4 significant figures, not 3.

Reviewer #2 (Remarks to the Author):

This manuscript describes a new synthetic method for the aryl alkyl ethers from aryl methyl sulfoxides with alcohols by K⁺O⁻t-Bu. A variety of ethers bearing functional groups could be prepared in good to excellent yields by simple protocol. Notably, as demonstrated in Scheme 4, even aminoalcohols were selectively reacted at OH group, which is useful for the synthesis of complex molecules. And the effect of counter cation was also discussed by DFT calculation.

The one of interesting points in this paper is chemoselectivity when using aminoalcohols. However, there is no explanation/discussion about that. Is it possible to discuss chemoselectivity by experiment or calculation? Some explanation would be necessary to show the importance of this work.

In addition, the effect of potassium cation and solvent is dramatic as shown in Table 1. I wonder that the difference in yield between KOMe and MeOH/K⁺O⁻t-Bu (entries 5 and 8). Is there possibility that more basic K⁺O⁻t-Bu deprotonates a C-H bond of methyl aryl sulfoxide to generate enolate like species as a key intermediate? If the formation of enolate is faster or more stable by DFT calculation, authors should consider this possibility. And the reaction using 2-naphthyl phenyl sulfoxide (no a C-H bond) might be good substrate to discuss that.

The metal-free syntheses of aryl alkyl ethers by use of K⁺O⁻t-Bu have been reported. For examples, Wang, Duan, and Zhao et al. has reported the synthesis of aryl alkyl ethers from anisole derivatives through nucleophilic aromatic substitution (S_NAr) (Org. Lett. 2018, 20, 4267-4272). In these reports, potassium cation was found to be critical and effect was discussed by DFT calculation. And same group has reported etherification of aryl sulfide (Org. Lett. 2018, 20, 4749-4753). These examples mean that K⁺O⁻t-Bu has already been demonstrated to promote similar S_NAr process.

In all, this manuscript definitely discloses the interesting reactivity of aryl sulfoxides, but major

concerns are lack of discussion/explanation and novelty. Therefore, I would not provide my recommendation for publication on Nature Communications. I think that it should be published in an organic chemistry journal.

Extra comments:

- 1) The authors used "cross coupling" in title. I feel awkward to this point because present reaction should go through nucleophilic aromatic substitution as proposed in Figure 1. Thus, the authors should revise title. Or "formal cross-coupling" may be correct...
- 2) I recommend the authors to use table style, not scheme style in scheme 5 and 6 because reaction conditions were similar.
- 3) 1,4-dioxane was selected to consider the solvent effect in DFT calculation. Why? Should the coordination of DME to potassium cation be considered in scheme 7C as well?

Reviewer #3 (Remarks to the Author):

The manuscript describes a C–O bond-forming reaction of aryl and heteroaryl methyl sulfoxides with alcohols. The reaction is promoted by potassium tert-butoxide and does not require a transition metal catalyst. Sulfoxides are typically not considered suitable electrophiles for cross-coupling reactions, and few examples of such reactions are described. Given that conversions of aryl sulfoxides to aryl ethers are unprecedented, the manuscript by Kozłowski and Jia is very noteworthy from the synthetic and mechanistic standpoints. The reaction takes place under relatively mild conditions and in the absence of transition metal catalysts. The scope of the reaction includes heteroaryl and aryl sulfoxides. These two features make the reaction potentially attractive for applications in medicinal chemistry and materials science. On the mechanistic side, the reaction presents an interesting problem, since sulfenate is a relatively poor leaving group, and the facility of the substitution as well as the potassium ion effects are remarkable. The authors present a mechanistic rationale supported by experimental and computational evidence that explains this reactivity. This is especially important in the context of N-heterocyclic ortho-sulfoxides for which sulfoxides were previously found to be more reactive in substitution reactions with alkoxides than bromides and chlorides (see for example, *Tetrahedron Letters*, 1983, 24, 3243-3246 and *Perkin Transactions 1*, 1984, 8, 1839-1845), and the reason for this enhanced reactivity remained unknown. The mechanism proposed by Kozłowski and Jia with the initial attack of the alkoxide at the sulfur may account for these observations in addition to explaining the reactivity with aryl sulfoxides and may have broader implications for reactions of other organosulfur compounds (sulfones, sulfolenes, etc.). All in all, I think that the paper describes chemistry that is broadly interesting and important and is suitable for publication in *Nature Communications*.

Additional comments:

1) For N-heterocyclic sulfoxides with the sulfoxide in the 2 and 4 positions vs N, do authors expect potassium cation to interact with the pi-system or the nitrogen atom? It seems that the latter option provides a stronger interaction. Some discussion of these effects may be in order, given the prior literature observations (see above).

2) In Scheme 4, "3 linear longest steps" and "5 linear steps" should be "3 (or 5) steps in the longest linear sequence".

We thank the reviewer for the detailed and constructive comments. We have revised the manuscript according to the suggestions of the reviewers. The changes made to the original manuscript and Supplementary Information have been documented by with yellow highlights.

Reviewer 1: I recommend the acceptance of this manuscript to Nature Communications.

Comments:

Aryl sulfides could act as nucleophiles in transition-metal-catalyzed cross coupling reactions, such as Liebeskind-Srogl reactions (ref. 22-25 in the manuscript), which usually proceed via two-electron transition-metal addition pathway. Aryl sulfones, on the other hand, could be good radical precursors, as shown in Julia fragmentation. Therefore, it is a very interesting question to probe the role of aryl sulfoxides in coupling reactions. It is intriguing to a broad range of chemists that aryl sulfoxides could react in a completely different pathway, especially independent of transition-metal catalysts. Jia, Kozlowski and coworkers developed an amazing cross-coupling reaction between aryl sulfoxides and alcohols. Remarkably, the protocol performed well undergo transition-metal-free manner in the presence of KO^tBu. Thus I am both excited and intrigued by this unprecedented and novel transformation. Moreover, the substrate scope is well demonstrated, and the compounds are well characterized. They also applied their method in the preparation of two bioactive molecules to demonstrate its synthetic potential. The novel mechanism disclosed in the manuscript, which was demonstrated by DFT calculation, certainly could lead to further developments. Jia and Kozlowski's hypothesis-driven research style as shown in the manuscript were inspired by JC Martin's work in the 70th, and applied the key intermediate in the modern organic methodology. Therefore, I recommend the acceptance of this manuscript to Nature Communications after address the following notes:

1. In manuscript, in conclusion part, it should be "late-stage", not "late-state".

Our reply: Following Reviewer 1's suggestion, we have changed "late-state" to "late-stage".

2. In SI, calculated value of compound 3h for HRMS should have 4 significant figures, not 3.

Our reply: We really appreciate Reviewer 1's careful reading. The calculated molecular value of compound 3h in SI was corrected to 293.1900.

Reviewer 2: I would not provide my recommendation for publication on Nature Communications. I think that it should be published in an organic chemistry journal.

3. This manuscript describes a new synthetic method for the aryl alkyl ethers from aryl methyl sulfoxides with alcohols by KO^tBu. A variety of ethers bearing functional groups could be prepared in good to excellent yields by simple protocol. Notably, as demonstrated in Scheme 4, even aminoalcohols were selectively reacted at OH group, which is useful for the synthesis of complex molecules. And the effect of counter cation was also discussed by DFT calculation. The one of interesting points in this paper is chemoselectivity when using aminoalcohols. However, there is no explanation/discussion about that. Is it possible to discuss chemoselectivity by experiment or calculation? Some explanation would be necessary to show the importance of this work.

Our reply: Thank you for pointing out the importance of the chemoselectivity of our protocol when using aminoalcohols as nucleophiles in the transition-metal free coupling strategy of

sulfoxides and alcohols. The mechanism of our protocol (Figure 1 in the manuscript for details) proceeds via initial nucleophilic attack of an alkoxide onto the S center of the sulfoxide which is facilitated by coordination to the aromatic substrate. Therefore, nucleophilicity of attacking group plays a very important role. In the presence of KOtBu as base in the reaction, alcohols are easily deprotonated to form alkoxide anions. However, amines would not be deprotonated by KOtBu since HOtBu is much more acidic than primary or secondary alkyl amines. From Bordwell, the pKa value in DMSO of ^tBuOH is 32.2, while that of Me₂NH is about 44 (see Bordwell, F. G. et al. J. Am. Chem. Soc. 1975, 97, 7006; Bordwell, F. G. J. Org. Chem. 1980, 45, 3295). For further reference, two comprehensive tables can be found at: http://evans.rc.fas.harvard.edu/pdf/evans_pKa_table.pdf; <https://www.chem.wisc.edu/areas/reich/pkatable/index.htm>). Therefore, the actual nucleophiles in the reaction system are an alkoxide anion and an amine when an aminoalcohol is employed as the coupling partner. Alkoxides anions are regarded as better nucleophiles than neutral amines accounting for why ethers are obtained as the sole products in a highly chemoselective fashion. Notably, the amine is retained intact allowing for further transformations. Leveraging this difference, opioid delta receptor agonist 7c was prepared in only three steps with good yield, and no protecting group was needed (Scheme 4)!

4. In addition, the effect of potassium cation and solvent is dramatic as shown in Table 1. I wonder that the difference in yield between KOMe and MeOH/ KO^tBu (entries 5 and 8). Is there possibility that more basic KO^tBu deprotonates a C–H bond of methyl aryl sulfoxide to generate enolate like species as a key intermediate? If the formation of enolate is faster or more stable by DFT calculation, authors should consider this possibility. And the reaction using 2-naphthyl phenyl sulfoxide (no a C–H bond) might be good substrate to discuss that.

Our reply: The reviewer has a good point. The pKa values of ^tBuOH and phenyl methyl sulfoxide are close with that of the latter being somewhat higher in DMSO. Since these effects are highly solvent dependent, we conducted a deprotonation experiment using 2-naphthyl methyl sulfoxide and benzyl chloride as a trapping agent. Using the same reaction conditions but without MeOH (KO^tBu in DME as 110 °C for 12 h, see Figure 1 below), no desired trapping product from deprotonation and alkylation with benzyl chloride was observed (by TLC, ¹H NMR and LCMS). On this basis, it appears that deprotonation of the α-proton of phenyl methyl sulfoxide is not competitive under these conditions.

Figure 1. Trapping experiment between 2-naphthyl methyl sulfoxide and benzyl chloride.

5. The metal-free syntheses of aryl alkyl ethers by use of KOt-Bu have been reported. For examples, Wang, Duan, and Zhao et al. has reported the synthesis of aryl alkyl ethers from anisole derivatives through nucleophilic aromatic substitution (S_NAr) (Org. Lett. 2018, 20, 4267-4272). In these reports, potassium cation was found to be critical and effect was discussed by DFT calculation. And same group has reported etherification of aryl sulfide (Org. Lett. 2018, 20, 4749-4753). These examples mean that KOt-Bu has already been demonstrated to promote similar S_NAr process.

Our reply: After we carefully read two papers mentioned (Org. Lett. 2018, 20, 4267-4272; Org. Lett. 2018, 20, 4749-4753), we respectfully disagree with the conclusion of Reviewer 2 that “that KOt-Bu has already been demonstrated to promote similar S_NAr process”.

Org. Lett. 2018, 20, 4749-4753 describes the formation of aryl alkyl ethers from ortho-thiomethylaryl nitriles and alcohols via a conventional S_NAr pathway (conjugate type addition to the aryl nitrile and then elimination of the sulfide). Org. Lett. 2018, 20, 4267-4272 outlines a similar strategy with ortho- and para-methoxyaryl nitriles. Further, these papers give no example of any aryl sulfoxides.

Both of these Org. Lett. cases rely on the presence of a separate activating group (nitrile) and leaving group (thiomethyl or methoxy). In our report, we directly displace the sulfoxide without any additional activating group. Since our method proceeds through a completely different and unprecedented pathway, a separate activating group is not required. Furthermore, our protocol could successfully tolerate electron-donating substituents, such as methoxyl (4d), methylthiol (4e), and vinyl (4g), which is different behavior from conventional S_NAr mechanisms.

All of the reviewers comment positively on the novelty. As Reviewer 2 stated above “This manuscript describes a new synthetic method for the aryl alkyl ethers from aryl methyl sulfoxides with alcohols by KO^tBu.” Reviewer 1 also pointed out that “Aryl sulfides could act as nucleophiles in transition-metal-catalyzed cross coupling reactions, such as Liebeskind-Srogl reactions (ref. 22-25 in the manuscript), which usually proceed via two-electron transition-metal addition pathway. Aryl sulfones, on the other hand, could be good radical precursors, as shown in Julia fragmentation. Therefore, it is a very interesting question to probe the role of aryl sulfoxides in coupling reactions”. Reviewer 3 adds that “Given that conversions of aryl sulfoxides to aryl ethers are unprecedented, the manuscript by Kozlowski and Jia is very noteworthy from the synthetic and mechanistic standpoints.”

Extra comments:

6. The authors used “cross coupling” in title. I feel awkward to this point because present reaction should go through nucleophilic aromatic substitution as proposed in Figure 1. Thus, the authors should revise title. Or “formal cross-coupling” may be correct...

Our reply: From Figure 1, the mechanism of our protocol is not conventional nucleophilic aromatic substitution, but via a nucleophilic attack of alkoxide to S center of the sulfoxide via coordinating to the aromatic substrate, generating a key S-tetravalent dialkoxyl aryl intermediate, which facilitates the subsequent rearrangement to yield the alkyl aryl ethers as the final product. In sharp contrast, the classic nucleophilic aromatic substitution is via the Meisenheimer intermediate (see Figure 2 below showing a para-substituted substrate as example). Therefore, the mechanism we proposed in the manuscript is clearly different from nucleophilic aromatic substitution. That said, we appreciate the suggestion of Reviewer 2, and changed our title to “formal cross-coupling”.

Nucleophilic aromatic substitution reaction (S_NAr)
para-substituted for instance (*ortho* is also favored)

Figure 2. Classic mechanism of nucleophilic aromatic substitution via a Meisenheimer intermediate.

7. I recommend the authors to use table style, not scheme style in scheme 5 and 6 because reaction conditions were similar.

Our reply: Following this suggestion of Reviewer 2, Scheme 5 and 6 have been formatted into table style and the titles have been changed to "Table" - see yellow highlights.

8. 1,4-dioxane was selected to consider the solvent effect in DFT calculation. Why?

Our reply: Dioxane, which has similar polarizability, was selected as a stand-in for DME, which is not available in Gaussian.

Should the coordination of DME to potassium cation be considered in scheme 7C as well?

Our reply: The DME coordination has been added to Scheme 7C (now Scheme 5C)

Reviewer 3: I think that the paper describes chemistry that is broadly interesting and important and is suitable for publication in Nature Communications.

The manuscript describes a C–O bond-forming reaction of aryl and heteroaryl methyl sulfoxides with alcohols. The reaction is promoted by potassium tert-butoxide and does not require a transition metal catalyst. Sulfoxides are typically not considered suitable electrophiles for cross-coupling reactions, and few examples of such reactions are described. Given that conversions of aryl sulfoxides to aryl ethers are unprecedented, the manuscript by Kozłowski and Jia is very noteworthy from the synthetic and mechanistic standpoints. The reaction takes place under relatively mild conditions and in the absence of transition metal catalysts. The scope of the reaction includes heteroaryl and aryl sulfoxides. These two features make the reaction potentially attractive for applications in medicinal chemistry and materials science. On the mechanistic side, the reaction presents an interesting problem, since sulfenate is a relatively poor leaving group, and the facility of the substitution as well as the potassium ion effects are remarkable. The authors present a mechanistic rationale supported by experimental and computational evidence that explains this reactivity. This is especially important in the context of N-heterocyclic ortho-sulfoxides for which sulfoxides were previously found to be more reactive in substitution reactions with alkoxides than bromides and chlorides (see for example, Tetrahedron Letters, 1983, 24, 3243-3246 and Perkin Transactions 1, 1984, 8, 1839-1845), and the reason for this enhanced reactivity remained unknown. The mechanism proposed by Kozłowski and Jia with the

initial attack of the alkoxide at the sulfur may account for these observations in addition to explaining the reactivity with aryl sulfoxides and may have broader implications for reactions of other organosulfur compounds (sulfones, sulfolenes, etc.). All in all, I think that the paper describes chemistry that is broadly interesting and important and is suitable for publication in Nature Communications.

Additional comments:

9. For N-heterocyclic sulfoxides with the sulfoxide in the 2 and 4 positions vs N, do authors expect potassium cation to interact with the pi-system or the nitrogen atom? It seems that the latter option provides a stronger interaction. Some discussion of these effects may be in order, given the prior literature observations (see above).

Our reply: It is entirely plausible that the nitrogen of the different heterocycles could coordinate the potassium ion. However, oxygen coordination is typically stronger than nitrogen coordination for potassium, so the sulfoxide oxygen would be expected to dominate. That said, it is entirely reasonable that substrates with a nearby nitrogen (4i, 4j, 4m, 4n, 4o, 4u) would benefit from bidentate coordination via the nitrogen and the sulfoxide to the potassium. For cases with an nitrogen distal to the sulfoxide (4k, 4l, 4p, 4q, 4r, 4s, 4t) it is not clear that the same benefits would follow. All told, there does not appear to be a strong relationship between the presence/absence of nearby nitrogen with yield, and even electron-withdrawing groups do not have a consistent effect (compare 4a/4b vs 4i/4j).

The following text has been added: “For heterocycles with a nitrogen adjacent to the sulfoxide (4i, 4j, 4m, 4n, 4o, 4u), reactivity may be further assisted by bidentate coordination of the potassium to the heterocyclic nitrogen and the sulfoxide oxygen. However, the lack of any distinct trends in the outcomes for these substrates relative to heterocycles with more distal nitrogens (4k, 4l, 4p, 4q, 4r, 4s, 4t) indicates that this effect does not predominate.”

10. In Scheme 4, “3 linear longest steps” and “5 linear steps” should be “3 (or 5) steps in the longest linear sequence”.

Our reply: Following Reviewer 3’s suggestion, in Scheme 4, “3 linear longest steps” and “5 linear steps” have been changed to “3 (or 5) steps in the longest linear sequence”, which has been highlighted in yellow.

Thank you for your consideration of this manuscript. The comments of the reviewers have helped us strengthen the manuscript to improve its impact and rigor. I hope, with the improvements that we have made, the manuscript is suitable for publication in Nature Communications.

REVIEWERS' COMMENTS:

Reviewer #2 (Remarks to the Author):

The authors correctly revised this manuscript with explanation as reviewers suggested. Thus, I recommend publication in Nature Communications.

Reviewer #3 (Remarks to the Author):

This is a review of the resubmission by Kozlowski and Jia. The authors have adequately addressed the critiques raised in the first round. I think the manuscript describes chemistry that is distinct from previously published alkoxide-mediated examples of S_NAr substitutions in electron-deficient substrate series. It nicely highlights the growing utility of organosulfur compounds in carbon-heteroatom bond-forming processes and the importance of counterion effects. I think the manuscript can be accepted in the present format, after the following minor issues have been addressed:

In Scheme 5:

Reaction A: $SOMe$ over the second reaction arrow should have a minus (anion) sign'

Reaction B: Following the pattern in the other two reactions, there should probably be " $-KOSMe$ " over the second reaction arrow.

Reviewer 2: I recommend publication in Nature Communications.

Comments:

The authors correctly revised this manuscript with explanation as reviewers suggested. Thus, I recommend publication in Nature Communications.

Reviewer 3: I think the manuscript can be accepted in the present format, after the following minor issues have been addressed.

Comments:

This is a review of the resubmission by Kozłowski and Jia. The authors have adequately addressed the critiques raised in the first round. I think the manuscript describes chemistry that is distinct from previously published alkoxide-mediated examples of $\text{S}_{\text{N}}\text{Ar}$ substitutions in electron-deficient substrate series. It nicely highlights the growing utility of organosulfur compounds in carbon-heteroatom bond-forming processes and the importance of counterion effects. I think the manuscript can be accepted in the present format, after the following minor issues have been addressed:

1. In Scheme 5: Reaction A: SOMe over the second reaction arrow should have a minus (anion) sign. Reaction B: Following the pattern in the other two reactions, there should probably be " $-\text{KOSMe}$ " over the second reaction arrow.

Our reply: Following the suggestions of Reviewer 3, in Scheme 5 (Fig. 5 in the revised manuscript), a minus (anion) sign has been added in Reaction A. In Reaction B, " $-\text{KOSMe}$ " has been added over the second reaction arrow. The changes have been highlighted in yellow.

Thank you for your consideration of this manuscript. The comments of the reviewers have helped us strengthen the manuscript to improve its impact and rigor. I hope, with the improvements that we have made, the manuscript is suitable for publication in Nature Communications.